# Protein arginine deiminase 4 antagonizes methylglyoxal-induced histone glycation

Qingfei Zheng [1], Adewola Osunsade [1,2] & Yael David [1,2,3,4]✉

Protein arginine deiminase 4 (PAD4) facilitates the post-translational citrullination of the core histones H3 and H4. While the precise epigenetic function of this modification has not been resolved, it has been shown to associate with general chromatin decompaction and compete with arginine methylation. Recently, we found that histones are subjected to methylglyoxal (MGO)-induced glycation on nucleophilic side chains, particularly arginines, under metabolic stress conditions. These non-enzymatic adducts change chromatin architecture and the epigenetic landscape by competing with enzymatic modifications, as well as changing the overall biophysical properties of the fiber. Here, we report that PAD4 antagonizes histone MGO-glycation by protecting the reactive arginine sites, as well as by converting already-glycated arginine residues into citrulline. Moreover, we show that similar to the deglycase DJ-1, PAD4 is overexpressed and histone citrullination is upregulated in breast cancer tumors, suggesting an additional mechanistic link to PAD4's oncogenic properties.

[1] Chemical Biology Program, Memorial Sloan Kettering Cancer Center, New York, NY 10065, USA. [2] Tri-Institutional PhD Program in Chemical Biology, New York, NY 10065, USA. [3] Department of Pharmacology, Weill Cornell Medicine, New York, NY 10065, USA. [4] Department of Physiology, Biophysics and Systems Biology, Weill Cornell Medicine, New York, NY 10065, USA. ✉email: davidshy@mskcc.org

In eukaryotes, nucleosomes are the fundamental unit of chromatin, composed of DNA and histone proteins[1]. Post-translational modifications (PTMs) on histones, including acetylation and methylation, are important in regulating chromatin structure and function during replication, transcription, and DNA damage[2,3]. It is speculated that the combinations of specific PTMs form a so-called "histone code"[4] that is established through a network of crosstalks between enzymatic modifications and determines the transcriptional state of a specific genomic locus[5,6]. Citrullination, which occurs on arginine residues and involves the deimination of the guanidino group, reduces the net charge of the side chain and was generally shown to promote chromatin fiber decompaction[7], although specific sites have also been associated with gene repression[8]. It was previously demonstrated that histone H3 arginine citrullination antagonizes methylation at the same residues by both blocking the modification sites and preventing the recruitment of the methyl-transferases[9,10]. Histone arginine citrullination is executed by the calcium-dependent enzyme, protein arginine deiminase 4 (PAD4). PAD4 substrates include specific sites on both core and linker histones[11], and it was suggested to play a role in determining cellular pluripotency as well as in the DNA damage response[12]. Although *PAD4* (*PADI4*) is a documented onco-gene[13–16], the regulatory function of histone citrullination in pathological and physiological processes is still poorly understood[12].

Whereas the most well-characterized histone modifications are enzymatic, in the past few years it has become clear that histones are also prime substrates of nonenzymatic covalent modifications that induce changes in chromatin structure and function[17]. We recently found that core histones are subjected to MGO glycation and that these adducts change chromatin architecture, the epigenetic landscape, and transcription, and particularly accumulate in disease states[18,19]. We determined that short or low-concentration exposure to MGO induces chromatin decompaction by compromising the electrostatic interactions of the histone tails with DNA, in a mechanism similar to acetylation. These initial adducts can rearrange and undergo crosslinking, which ultimately leads to chromatin fiber hyper-compaction. While MGO reacts with histones nonenzymatically, we found that DJ-1/PARK7 is a potent histone deglycase, preventing the accumulation of histone glycation in vitro and in cells[18]. Since MGO rapidly reacts with the guanidino group of arginines[20,21], we tested the effect of MGO glycation on cellular H3R8 methylation mark and found that MGO induces a reduction in H3R8me2 levels. As multiple arginine residues on histones, including H3R8, were shown to be substrates of PAD4[12], we hypothesized that citrullination, which alters the charge of the amino-acid side chains and reduces their reactivity against electrophiles, blocks MGO glycation on arginines and vice versa. In this study, we address this hypothesis and find that beyond the direct competition between citrullination and MGO glycation, PAD4 itself acts as a deglycase, mediating the conversion of arginine glycation adducts into citrulline (Fig. 1).

## Results

### PAD4-mediated citrullination prevent histone MGO glycation.
First, we aimed to analyze the biochemical mechanism governing the crosstalk between histone citrullination and glycation. To do so, we purified recombinant PAD4 (Supplementary Fig. 1) and tested its activity in vitro on a range of substrates with increasing complexities, including free histone H3, nucleosome core particles (NCPs), and homododecameric (12-mer) nucleosomal arrays, which mimic the minimal chromatin fold[22]. Our results indicate that PAD4 has increased reactivity towards more physiological substrates, with highest detected activity on 12-mer arrays (Supplementary Fig. 2), in a $Ca^{2+}$-dependent manner (Supplementary Fig. 3). Next, we utilized unmodified and PAD4-citrullinated NCPs as substrates of MGO glycation to test the direct competition between these modifications. Our results indicate that citrullination protects NCPs from undergoing MGO glycation, since after a 12-h MGO treatment the citrullinated NCPs were substantially less glycated compared to unmodified NCPs (Fig. 2a). Moreover, this protective effect of PAD4-mediated citrullination against MGO glycation is dose dependent (Supplementary Fig. 4).

### PAD4 converts MGO-glycated histone arginines to citrullines.
Next, we tested the reciprocal competition—that is, whether MGO glycation protects NCPs from undergoing PAD4-mediated citrullination. To do so, we pretreated NCPs with MGO and, after removing the excess MGO, used the glycated-NCPs as substrates for PAD4 enzymatic reaction. The results unexpectedly indicated that PAD4 is still able to modify the glycated nucleosomes and that the citrullination is added at the expense of glycation, as evident by the decrease in MGO glycation and increase in citrullination signals (Fig. 2b). To test this newly identified histone deglycation activity of PAD4 and compare it to DJ-1, which we recently identified as a histone deglycase[18], we either added the enzymes to NCPs concurrently with MGO (C), or after a short (S) or a long (L) treatment with MGO (in both cases unreacted MGO was removed prior to enzymes addition). The results indicate that DJ-1 is capable of removing MGO glycation from NCPs when added concurrently or after a short exposure to MGO. However, glycation was persistent following a long exposure, which allows the rearrangement of the glycation adducts into late-stage products (Figs. 1 and 2b). In contrast to DJ-1 and the no-enzyme control, PAD4 was able to remove MGO-adducts and install citrulline, regardless of the incubation time (Fig. 2b), suggesting it is active on both early- (aminocarbinol) and late-stage (carboxyethyl arginine, CEA) MGO adducts. Indeed, applying both DJ-1 and PAD4 resulted in the almost complete abolishment of the glycation (Fig. 2b). To further validate the direct deglycase activity of PAD4, we performed a deglycation assay on biotinylated H3 and H4 N-terminal peptide substrates. The peptides were incubated with MGO, immobilized, washed and finally treated with PAD4. The reactions were analyzed by both dot blot (Fig. 2d) and LC-MS (Fig. 2e, Supplementary Figs. 5 and 6), confirming that PAD4 directly removes MGO-glycation adducts from both H3 and H4 tails in vitro. In contrast, a similar histone arginine modifying enzyme, protein arginine N-methyltransferase 1 (PRMT1), was not capable of converting the MGO-glycated H4-R3 to methylated arginine (Supplementary Fig. 7). Finally, to examine the effect of PAD4 activity on chromatin compaction, we performed a $Mg^{2+}$ precipitation analysis on 12-mer arrays substrate. This assay relies on magnesium inducing the aggregation and precipitation of the 12-mer fibers, so the less compacted the arrays are, the more $Mg^{2+}$ is required to precipitate them. Our results demonstrate that histone citrullination decompacts the chromatin arrays to a lesser degree than MGO treatment, and that PAD4 rescues the majority of MGO-dependent chromatin decompaction for all incubation conditions (Fig. 2c). This is in contrast to the deglycase DJ-1[23–25], which rescues glycation-induced decompaction only under short or co-incubation conditions (Supplementary Fig. 8).

### PAD4 antagonizes histone MGO glycation.
To investigate PAD4-mediated crosstalk between glycation and citrullination in native chromatin, we examined the impact of its expression on histone arginine citrullination, glycation and methylation in cells.

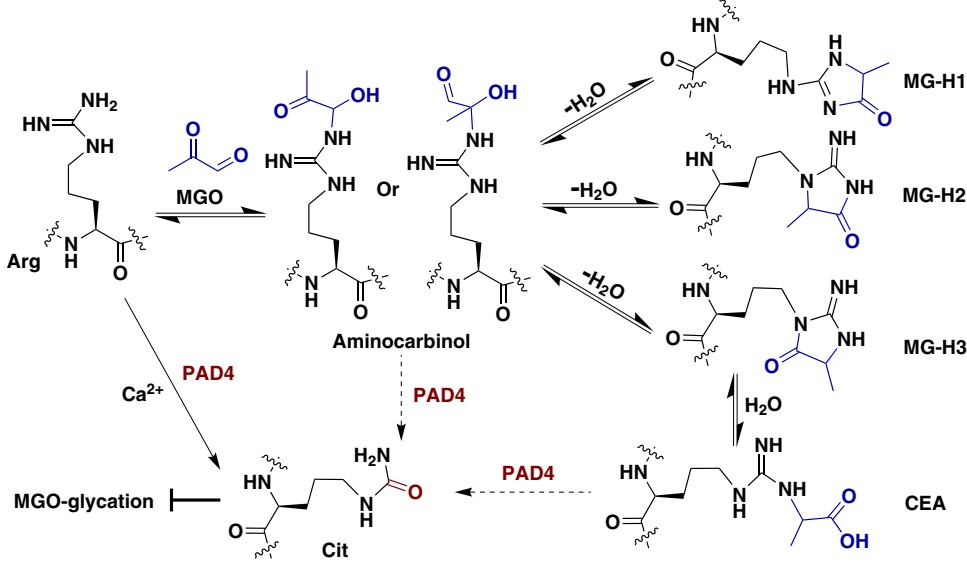

**Fig. 1 PAD4 antagonizes methylglyoxal-induced histone glycation.** Overall schematic showing MGO-induced histone glycation and citrullination/deglycation activity of PAD4.

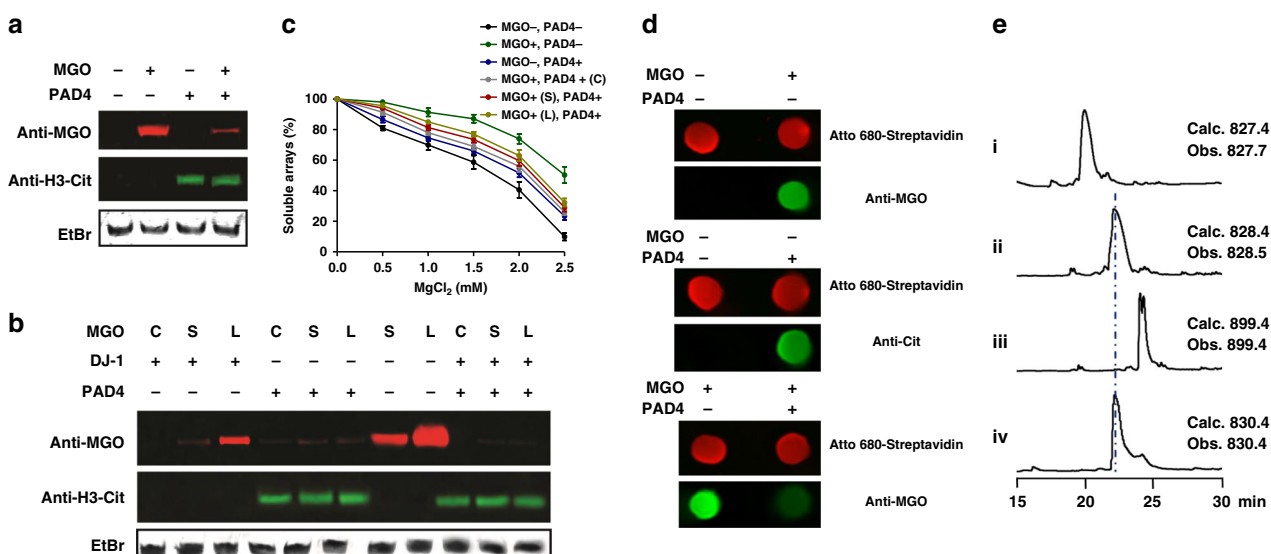

**Fig. 2 In vitro citrullination and glycation assays. a** NCPs were first citrullinated by PAD4 followed by a 5 mM MGO treatment at 37 °C overnight. The reactions were separated on a native gel followed by a western blot analysis. (**b**) NCPs were pretreated with 5 mM MGO for short (S, 6 h) or long (L, overnight) periods and then incubated with PAD4, or concurrently (C) incubated with both the enzyme and MGO for the same time. **c** PAD4 was incubated with glycated nucleosomal arrays after short (S) or long (L) treatment with MGO, or co-incubation (C). Error bars represent the standard deviation from three different experiments. Data are presented as mean values ± SEM. **d** Dot blot analysis of MGO glycation and PAD4-mediated deglycation on H3 N-terminal peptide substrate. The H3 peptide sequence is NH$_2$-ARTKQTARKSTGGKAPRK(Bio)A-COONH$_2$, and the biotin signal (imaged by Atto 680-Streptavidin) was used as loading control. **e** LC-MS analysis of modified and unmodified H4 peptides ($\lambda = 214$ nm): (i) H4-R3(1-6) peptide, (ii) H4-Cit3(1-6) peptide, (iii) MGO-treated H4-R3(1-6), (iv) MGO-glycated H4-R3(1-6) followed by PAD4 treatment in H2$^{18}$O. The H4 synthesized peptide sequences are AcNH-SGRGK(Bio)G-COONH$_2$ and AcNH-SGCitGK(Bio)G-COONH$_2$.

As expected, PAD4 overexpression in 293 T cells induces an increase in histone citrullination at the expense of arginine methylation (Fig. 3a and Supplementary Fig. 9)[9,10]. In analogy to our in vitro results, treating these cells with increasing amounts of MGO induced the accumulation of H3 and H4 glycation (in addition to crosslinking), which was partially suppressed by PAD4 overexpression (Fig. 3a). To dissect the deglycation function of PAD4, we performed a pulse-chase experiment where cells were first pulsed with a gradient of MGO concentrations and then washed with fresh media. After a 6-h recovery, cells were transfected with PAD4 and grown for additional 12 h before final harvesting. Analysis of the histone samples from these cells revealed that overexpression of PAD4 resulted in a decreased MGO-glycation signal, suggesting PAD4 is actively removing MGO adducts from histones in live cells (Supplementary Fig. 10). In a complementary analysis, similar assays were performed with an alkynyl methylglyoxal probe and further confirmed the deglycation activity of PAD4[26]. Moreover, a global chromatin compaction analysis by micrococcal nuclease (MNase) digestion revealed that PAD4 rescues MGO-induced chromatin

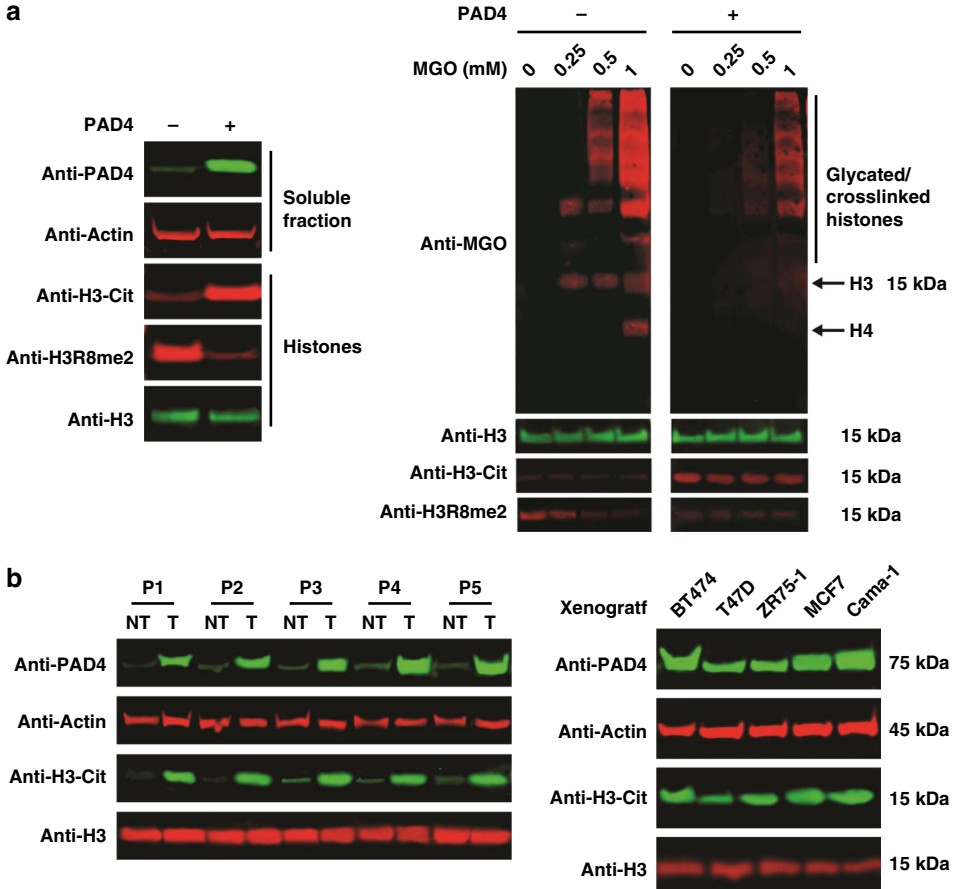

**Fig. 3 In vivo citrullination and glycation analyses. a** Left: western blot analysis of both soluble and histone fractions of wild-type and PAD4-overexpressing 293 T cells. Right: western blot analysis of the histone fraction taken from the same cells that were treated with increasing concentration of MGO. **b** Western blot analysis of clinical tumor (T) and nontumor (NT) samples from five different breast cancer patients (P1-P5) (left) or breast cancer xenografts (right).

decompaction (Supplementary Fig. 11). Finally, to demonstrate the direct binding of glycated histone tail to PAD4, we utilized a biotin-H3 peptide (residues 1–18)[18], either glycated or non-glycated, to demonstrate PAD4 binding regardless of its catalytic activity (Supplementary Fig. 12). Together, these results establish that PAD4 directly affects chromatin compaction by regulating the epigenetic crosstalk between histone MGO glycation, citrullination, and methylation.

**PAD4 is overexpressed in breast cancer**. Previous studies revealed the overexpression of PAD4 in a variety of cancers and suggested it could drive tumor pathogenesis through multiple potential mechanisms[12–16]. Since we recently reported high levels of both histone MGO glycation and DJ-1 expression in breast cancer[18], we utilized the same samples to analyze PAD4 and histone citrullination levels. As presented in Fig. 3b, all patient tumor samples display substantial overexpression of PAD4 and increased levels of histone citrullination. In addition, xenograft tumor samples show diverse levels of PAD4 expression that correlate with degree of histone citrullination. In support, 293 T cells treated with increasing amounts of MGO, which mimics the metabolic stress that exists in cancer cells, show better survival when overexpressing PAD4 but not the catalytically dead mutant C645S (Supplementary Fig. 13)[9,10]. To investigate the impact of PAD4 on histone PTM crosstalk in breast cancer cells, MCF7 cells were pretreated with the PAD4 inhibitor GSK484[27] followed by a treatment with increasing concentrations of MGO. The

analysis, presented in Supplementary Fig. 14, shows that inhibition of PAD4 increases histone glycation and arginine methylation. Together, these data suggest a role for PAD4 in cancer proliferation through the regulation of chromatin structure and function.

## Discussion

This study revealed a new crosstalk between histone glycation and citrullination, which is mediated by PAD4. Our data (Fig. 2 and Supplementary Fig. 15) support a dual-functional model whereby PAD4 is capable of not only protecting arginine via deamination but rewriting MGO-adduct intermediates into citrulline, which is currently speculated to be a terminal product (Fig. 1, Supplementary Figs. 16 and 17). Although it was reported that free citrullines can be converted to arginines by argininosuccinate synthetase (ASS) and argininosuccinate lyase (ASL) in the citrulline-NO cycle[28], to date there is no report of peptidyl citrulline being reverted back to arginine[12]. Therefore, citrullination is likely to accumulate in long lived proteins, such as histones, and prevent the target residues from undergoing electrophilic damage such as glycation[29]. Based on this new deglycation activity of PAD4, it could potentially play a regulatory role in rescuing nonenzymatic damage and downstream changes in chromatin structure and function.

There are several implications for this newly identified rewriting function of PAD4 on MGO-glycated histones. First, it provides an additional pathway for the repair of

pathophysiological histone glycation. We have previously shown that histone glycation is a modification that accumulates under metabolic stress, such as in highly-proliferating breast cancer tumors, and that it changes patterns of transcription and chromatin architecture. In that regard, both DJ-1 and PAD4 are proposed oncoproteins and targets of cancer therapy although the mechanisms are not fully understood[13–16,30]. Indeed, we have shown that breast cancer patient samples contain massive over-expression of both DJ-1[18] and PAD4 (Fig. 3). There are several efficient DJ-1 and PAD4 inhibitors reported, some of which are in pre-clinical trials[30–32], raising the potential of combinational therapy targeting these enzymes simultaneously.

Counterintuitively, our breast cancer patients' samples also have high levels of glycation, suggesting that the overexpression of DJ-1 and PAD4 is not sufficient to remove all the glycation adducts. The results we present here indicate that this could be due to the fact that DJ-1 and PAD4 also work through distinct catalytic mechanisms. While DJ-1 erases early MGO-glycation adducts from both lysines and arginines, PAD4 is only active on arginine residues, that are more reactive towards MGO relative to lysines[20,21]. However, PAD4 is superior due to its rewriting activity, that is, it removes the glycation adduct from arginine and protects it from further damage by converting it to citrulline. It is noteworthy that other PAD enzymes, as well as PAD4 itself, may target glycation adducts in other substrates although this remains to be determined[12]. We thus cannot exclude that additional repair mechanisms may exist to protect from or erase other glycation damage on histones or other cellular proteins.

Another important implication of this finding is the three-way metabolic crosstalk between glycation, citrullination, and methylation that compete for the same arginine sites on histones (Fig. 4). All these modifications contribute to chromatin architecture regulation. Both histone glycation at its early stages[18] and citrullination[7,11] induce chromatin decompaction. Indeed, the synergistic activity between PAD4 and DJ-1 leaves the treated glycated and non-glycated chromatin less compacted, with the newly added citrullination inducing chromatin fiber decompaction compared with untreated one (Figs. 2c, S8, and S11). These changes in the chromatin landscape directly correlate with the accessibility of the metabolites generated from the associated pathways: glycation with sugar glycolysis[33], citrullination with calcium homeostasis[12], and methylation with S-adenosyl methionine (SAM) metabolism[34], suggesting that this crosstalk is influenced by diet, metabolic state and the cellular micro-environment[35,36]. The balance between MGO, $Ca^{2+}$, and SAM can be regulated through multiple processes including endo-plasmic reticulum (ER) stress, reactive oxygen species (ROS), and

mTOR signaling, providing an additional potential link to changes in gene expression[37–39]. Together with our previous identification of the interrelationship between glycation and methylation[9,10], this work suggests a three-way crosstalk (Fig. 4) and new insights into the link between metabolism, epigenetics, and human disease[40–43].

## Conclusions

Metabolic syndromes and diabetes increase the risk for certain diseases such as cancer[44,45]. However, the mechanism behind this correlation is poorly understood. Methylglyoxal, a reactive dicarbonyl sugar metabolite found in cells under metabolic stress[19], can nonenzymatically modify arginine and lysine residues in histone proteins[20], making it a new epigenetic marker linking metabolism and disease. Histone MGO glycation induces changes in chromatin architecture and the epigenetic landscape, abrogating gene transcription[18]. In this study, we found that PAD4 exhibits a dual function of antagonizing histone MGO glycation by both removing glycation adducts from arginines and converting the unmodified side chains into citrulline, protecting them from undergoing glycation. PAD4 induces changes in chromatin architecture[46] by regulating not only charges of arginine residues but also the MGO-glycation levels on histones. This unique function together with the overexpression of PAD4 in breast tumors provide insights into a potential mechanism for its function in cancer cells and understandings of the correlation between metabolism and cancer epigenetics. Overall, these findings expand our understanding of PAD4 biochemistry and its pathophysiological function in human health[47–51].

## Methods

**General materials and methods**. UV spectroscopy was performed on NanoDrop 2000c (Thermo Scientific). Biochemicals and media were purchased from Fisher Scientific or Sigma–Aldrich Corporation unless otherwise stated. T4 DNA ligase, DNA polymerase, and restriction enzymes were obtained from New England BioLabs. PCR amplifications were performed on an Applied Biosystems Veriti Thermal Cycler using either Taq DNA polymerase (Vazyme Biotech) for routine genotype verification or Phanta Max Super-Fidelity DNA Polymerase (Vazyme Biotech) for high-fidelity amplification. Site-specific mutagenesis was performed according to the standard procedure of the QuickChange Site-Directed Mutagenesis Kit purchased from Stratagene (GE Healthcare) or Mut Express II (Vazyme Biotech). Primer synthesis and DNA sequencing were performed by Integrated DNA Technologies and Genewiz, respectively. PCR amplifications were performed on a Bio-Rad T100TM Thermal Cycler. Centrifugal filtration units were purchased from Millipore, and MINI dialysis units purchased from Pierce. Size exclusion chromatography was performed on an AKTA FPLC system from GE Healthcare equipped with a P-920 pump and UPC-900 monitor. Sephacryl S-200 columns were obtained from GE Healthcare. All the western blots were performed using the primary antibodies annotated in Supplementary Table 1 and fluorophore-labeled secondary antibodies annotated in Supplementary Table 2 following the protocol

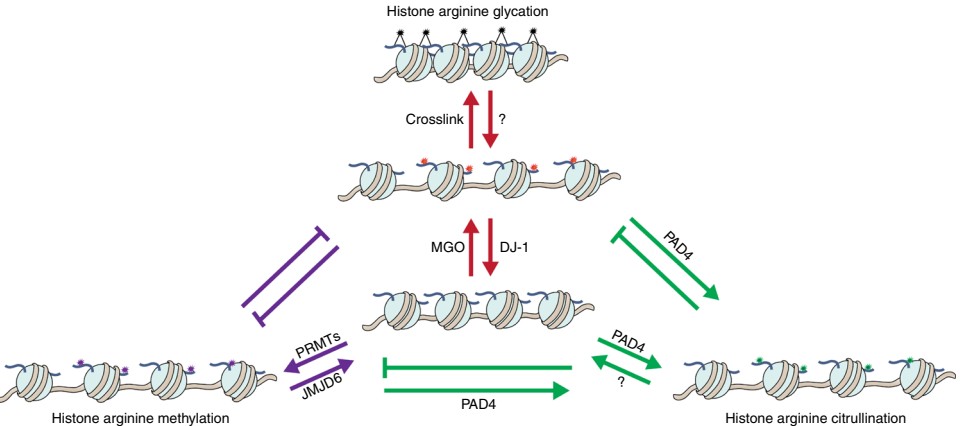

**Fig. 4 Glycation, citrullination, and methylation have an interaction network.** A proposed model showing the three-way crosstalk of histone glycation, citrullination, and methylation.

recommended by the manufacture. Blots were imaged on Odyssey CLx Imaging System (Li-Cor). Amino-acid derivatives and coupling reagents were purchased from AGTC Bioproducts. Dimethylformamide (DMF), dichloromethane (DCM), and triisopropylsilane (TIS) were purchased from Fisher Scientific and used without further purification. Hydroxybenzotriazole (HOBt) and O-(benzotriazol-1-yl)-N,N,N′,N′-tetramethyluronium hexafluorophosphate (HBTU) were purchased from Fisher Scientific. Trifluoroacetic acid (TFA) was purchased from Fisher Scientific. N,Ndiisopropylethylamine (DIPEA) was purchased from Fisher Scientific. Analytical reversed-phase HPLC (RP-HPLC) was performed on an Agilent 1200 series instrument with an Agilent C18 column (5 μm, 4 × 150 mm), employing 0.1% TFA in water (HPLC solvent A), and 90% acetonitrile, 0.1% TFA in water (HPLC solvent B) as the mobile phases. Analytical gradients were 0–70% HPLC buffer B over 45 min at a flow rate of 0.5 mL/min, unless stated otherwise. Preparative scale purifications were conducted on an Agilent LC system. An Agilent C18 preparative column (15–20 μm, 20 × 250 mm) or a semi-preparative column (12 μm, 10 mm × 250 mm) was employed at a flow rate of 20 mL/min or 4 mL/min, respectively. HPLC Electrospray ionization MS (HPLC-ESI-MS) analysis was performed on an Agilent 6120 Quadruple LC/MS spectrometer (Agilent Technologies). All the immunoblotting experiments in this research were done at least in triplets. The error bars in all the figures represent the standard deviation from three different experiments.

**Recombinant histone expression and purification**. Recombinant human histones H2A, H2B, H3.2, and H4 were expressed in E. coli BL21 (DE3) or E. coli C41 (DE3), extracted by guanidine hydrochloride and purified by flash reverse chromatography as previously described[52]. The purified histones were analyzed by RP-LC-ESI-MS.

**Preparation of histone octamer and '601' DNA**. Octamers were prepared as previously described[52]. Briefly, recombinant histones were dissolved in unfolding buffer (20 mM Tris-HCl, 6 M GdmCl, 0.5 mM DTT, pH 7.5), and combined with the following stoichiometry: 1.1 eq. H2A, 1.1 eq. H2B, 1 eq. H3.2, 1 eq. H4. The combined histone solution was adjusted to 1 mg/mL concentration transferred to a dialysis cassette with a 7000 Da molecular cutoff. Octamers were assembled by dialysis at 4 °C against 3 × 1 L of octamer refolding buffer (10 mM Tris-HCl, 2 M NaCl, 0.5 mM EDTA, 1 mM DTT, pH 7.5) and subsequently purified by size exclusion chromatography on a Superdex S-200 10/300 column. Fractions containing octamers were combined, concentrated, diluted with glycerol to a final 50% v/v and stored at −20 °C. The 147-bp 601 DNA fragment was prepared by digestion from a plasmid containing 30 copies of the desired sequence (flanked by blunt EcoRV sites on either site), and purified by PEG-6000 precipitation as described before[53].

**Mononucleosome assembly**. The mononucleosome assembly was performed according to the previously described salt dilution method with slight modification[54]. Briefly, the purified wild-type octamers were mixed together with 601 DNA (1:1 ratio) in a 2 M salt solution (10 mM Tris pH 7.5, 2 M NaCl, 1 mM EDTA, 1 mM DTT). After incubation at 37 °C for 15 min, the mixture was gradually diluted (9 × 15 min) at 30 °C by dilution buffer (10 mM Tris pH 7.5, 10 mM NaCl, 1 mM EDTA, 1 mM DTT). The assembled mononucleosomes were concentrated and characterized by native gel electrophoresis (5% acrylamide gel, 0.5 × TBE, 120 V, 40 min) using ethidium bromide (EtBr) staining.

**Nucleosomal array assembly**. Dodecameric repeats of the 601 sequence separated by 30-bp linkers were produced from pWM530 using EcoRV digestion and PEG-6000 precipitation according to the published procedure[55]. Homotypic dodecameric arrays were assembled from purified octamers and recombinant DNA in the presence of buffer DNA (MMTV) by salt gradient dialysis as previously described[56]. The resulting arrays were purified and concentrated using $Mg^{2+}$ precipitation at 4 °C[54].

**Expression of recombinant PAD4**. The pGEX-PAD4 plasmid was a kind gift from Prof. Paul Thompson (UMass Medical School). The GST-tagged PAD4 protein was expressed in E. coli Rosetta (DE3) cells with an overnight IPTG induction at 16 °C. The bacterial pellet was lysed by sonication and lysate cleared by centrifugation at 12,000 r.p.m. for 30 min. Lysate was loaded on GSTrap HP Column (GE Healthcare) and eluted on AKTA FPLC (GE Healthcare) by gradient L-glutathione (reduced, Sigma). The GST tag was cleaved by Precission Protease overnight during dialysis, and the cleaved proteins was purified by reverse GSTrap HP Column and size exclusion chromatography on AKTA FPLC. Purified recombinant proteins were analyzed by SDS-PAGE, and concentrated using stirred ultrafiltration cells (Millipore) according to the manufacturer's protocol. The concentration of each protein was determined using 280 nm wavelength on a NanoDrop 2000c (Thermo Scientific).

**Peptide synthesis**. Standard Fmoc-based Solid Phase Peptide Synthesis (FmocSPPS) was used for the synthesis of peptides in this study. The peptides were synthesized on ChemMatrix resins with Rink Amide to generate C-terminal

amides. Peptides were synthesized using manual addition of the reagents (using a stream of dry $N_2$ to agitate the reaction mixture). For amino-acid coupling, 5 eq. Fmoc protected amino acid were preactivated with 4.9 eq. HBTU, 5 eq. HOBt, and 10 eq. DIPEA in DMF and then reacted with the N-terminally deprotected peptidyl resin. Fmoc deprotection was performed in an excess of 20% (v/v) piperidine in DMF, and the deprotected peptidyl resin was washed with DMF to remove trace piperidine. Cleavage from the resin and side-chain deprotection were performed with 95% TFA, 2.5% TIS, and 2.5% $H_2O$ at room temperature for 1.5 h. The peptides were then precipitated with cold diethyl ether, isolated by centrifugation and dissolved in water with 0.1 % TFA followed by RP-HPLC and ESI-MS analyses. Preparative RP-HPLC was used to purify the peptides of interest.

**In vitro biochemical assays**. The PAD4 citrullination assays were performed in the buffer (pH 7.5) containing 50 mM Tris-HCl, 5 mM $CaCl_2$ and 2 mM DTT (freshly added). For free histone H3 citrullination, 50 μM H3 were treated with 5 μM PAD4 at 37 °C for 2 h, and were analyzed by sodium dodecyl sulfate polyacrylamide gel electrophoresis (SDS-PAGE) followed by western blot analysis. For nucleosome core particle (NCP) citrullination, 1 μM NCPs were treated with 0.1 μM PAD4 at 37 °C for 2 h. The citrullinated NCPs were analyzed by SDS-PAGE or native page electrophoresis followed by western blot analysis. Nucleosomal array citrullination assays were similarly prepared using 1 μM dodecameric arrays and 0.1 μM PAD4 with slight modification, that is, the concentration of $CaCl_2$ was reduced to 100 μM to prevent the arrays from precipitation.

For the NCP citrullination-glycation assays, the wild-type or citrullinated NCPs (1 μM) were treated with 5 mM MGO (Sigma) in 1 × PBS buffer (pH 7.4) at 37 °C for 6 h (short treatment) or overnight (long treatment)[18]. The buffer exchange between citrullination and glycation assays was performed using 0.5 mL Centrifugal Filter (3 K, Millipore) with a 120-fold v/v for the removal of MGO or $Ca^{2+}$ from the reaction buffer systems. The co-incubation of NCPs, MGO and PAD4 (or deglycase DJ-1) were also performed under the same conditions at 37 °C overnight. The modified NCPs were analyzed by SDS-PAGE (without boiling or lyophilizing the sample) or native gel followed by western blot analysis. For the SDS-PAGE analysis, H3 was used as loading control, while '601' DNA was used as loading control (by ethidium bromide staining) in the native gel analysis.

For peptide deglycation assays, 2 mM of the peptide substrate was incubated with 10 mM MGO in 1× PBS buffer (pH 7.4) at 37 °C for 30 min and then enriched by magnetic streptavidin beads (Thermo Fisher Scientific, 65602). After being washed by 1× PBS buffer, the glycated peptide was eluted with 100 mM glycine buffer (pH 2.5) and then dialyzed by using citrullination buffer (Tris-HCl), followed by the incubation with 100 μM PAD4 at 37 °C for 2 h. The elution and reaction buffers used in peptide deglycation were made with $H_2^{18}O$ (Sigma–Aldrich, 329878). The reactions were analyzed by dot blot and LC-MS.

For PRMT1 methylation assays, 50 μM full-length histone H4 was first incubated with (or without) 1 mM MGO in 1 × PBS buffer at 37 °C for 2 h, and then treated with human recombinant 5 μM PRMT1 heterologously expressed from Sf9 insect cells (Sigma, SRP0141) together with 100 μM SAM (Sigma) at 37 °C for 2 h. The products were separated by SDS-PAGE followed by western blot analysis using the corresponding antibodies.

**$MgCl_2$ precipitation of nucleosomal arrays**. The $MgCl_2$ precipitation of nucleosomal arrays was performed according to the published procedure[56]. Briefly, increasing concentrations of $MgCl_2$ were added to the nucleosomal arrays and the reaction was incubated on ice for 10 min, followed by a 10-min 17,000 rcf spin at 4 °C. The A260 of the supernatant was measured and used to evaluate the fraction of soluble arrays.

**Expression of PAD4 in 293 T cells**. The PAD4 gene was amplified from pGEX-PAD4 by PCR. The primers used are 5′-ATATGCGGCCGCTACCCATACGAT GTTCCAGATTACGCTATGGCCCAGGGGACATTG-3′ (NotI) and 5′-ATATG GTACCTCAGGGCACCATGTTCCACC-3′ (KpnI). The purified PCR product was inserted into pCMV plasmid by restriction endonuclease cloning. The catalytically dead mutant PAD4-C645S plasmid was constructed by site-directed mutagenesis using 5′-GGAGGTGCACAGCGGCACCAACG-3′ and 5′-CGTTGGTGCCGCTG TGCACCTCC-3′ as primers. The HA-tagged PAD4 was overexpressed in HEK 293 T cells using Lipofectamine 2000 Transfection Reagent (Thermo Fisher Scientific) according to the manufacturer's protocol. HEK 293 T cells (ATCC) were cultured at 37 °C with 5% $CO_2$ in DMEM medium supplemented with 10% fetal bovine serum (FBS) (Sigma–Aldrich), 2 mM L-glutamine and 500 units mL$^{-1}$ penicillin and streptomycin. Cells were stimulated with 2 μM calcium ionophore (Sigma–Aldrich, A23187) for 60 min at 37 °C before lysis, and then the PAD4 expression was detected by western blot analysis with anti-PAD4 and anti-HA antibodies.

**Immunoprecipitation and pull down**. The glycated and unglycated N-terminal biotinylated H3 peptides (residues 1–18) were used in the IP assays as previously described[18]. The glycated peptides were prepared by a treatment of MGO in a 1:3 (peptide: MGO) stoichiometry at 37 °C for 30 min as described above. Recombinant PAD4-C645S or total 293 T cell lysate containing overexpressed HA-PAD4-C645S were incubated with the peptides in 4 °C for 2 h after which the peptide was

pulled down by BSA-blocked Streptavidin Magnetic Beads (Thermo Scientific). Next, beads were washed three times with 1× PBS buffer (pH 7.4), boiled, separated on SDS-PAGE and analyzed by western blot with anti-PAD4 or anti-HA.

**Salt extraction of histones from cells**. The extraction of histones from cells was performed according to the previously described high-salt extraction method[57]. Briefly, the cell lysis solution was prepared using extraction buffer (10 mM HEPES pH 7.9, 10 mM KCl, 1.5 mM $MgCl_2$, 0.34 M sucrose, 10% glycerol, 0.2% NP40, protease and phosphatase inhibitors to 1× from stock). After spinning down, the pellet was extracted using a no-salt buffer (3 mM EDTA, 0.2 mM EGTA). After discarding the supernatant, the final pellet was extracted by using high-salt buffer (50 mM Tris pH 8.0, 2.5 M NaCl, 0.05% NP40) in 4 °C cold room for 1 h. After spinning down, the supernatant containing extracted histones was collected for further analyses.

**Tumor samples**. All cell-culture reagents were obtained from Thermo Fisher Scientific unless otherwise indicated. The cell lines for tumor xenografts were maintained at 37 °C and 5% $CO_2$ in humidified atmosphere. MCF7 was obtained from DSMZ, while T47D, BT474, ZR75-1, and Cama-1 cell lines used in this study were all sourced from ATCC. The cells were grown in DMEM/F12 supplemented with 10% FBS, 100 μg/mL penicillin, 100 mg/mL streptomycin, and 4 mmol/L-glutamine. All the cell lines tested negative for Mycoplasma and authenticated by short-tandem repeat (STR) analysis. Six-to-eight-week-old nu/nu athymic BALB/c female mice were obtained from Harlan Laboratories, Inc., and maintained in pressurized ventilated caging. All the studies were performed in compliance with institutional guidelines under an Institutional Animal Care and Use Committee-approved protocol (MSKCC#12–10–016). Xenograft tumors were established in nude mice by subcutaneously implanting 0.18-mg sustained release 17β-estradiol pellets with a 10 g trocar into one flank followed by injecting $1 \times 10^7$ cells suspended 1:1 (volume) with reconstituted basement membrane (Matrigel, Collaborative Research) on the opposite side 3 days afterward.

The clinical samples (MSKCC set) used in this study were obtained from the Biobank of MSKCC. The patients with breast cancer and either recurrence of disease after receiving adjuvant therapy or WHO-defined progression of metastatic disease on therapy were prospectively enrolled on an IRB approved tissue collection protocol (IRB#06-163). Informed consent was obtained from all patients. All patients underwent biopsy of at least a single site to document progressive disease. Mutational analysis of the metastatic biopsy was performed on fresh frozen specimens. Formalin fixed paraffin embedded (FFPE) blocks of the pretreatment primary tumor was obtained where possible for comparison. The presence of tumor, in both frozen samples and FFPE tissue sections, was confirmed by the study pathologist. Western blot analyses of PAD4 expression and histone citrullination were performed on fresh frozen specimens.

**Cell fractionation**. The cytosolic and nuclear fractions were prepared using NEPER Nuclear and Cytoplasmic Extraction Reagents (Thermo Scientific) according to the manufacturer's protocol. Histones were extracted from the pellet using high-salt extraction protocol as described above[7]. Histone extraction from tumor xenografts and patient samples followed a similar protocol, with slight variation, where tumors homogenized by mild sonication prior to extraction. Purity of fractionation was evaluated using the following antibodies: anti-Actin (cytosol), anti-MEK ½ (nucleoplasm) and anti-H3 (chromatin).

**Pulse-chase experiments**. 293 T cells were treated with a gradient of MGO for 12 h before the medium was changed to MGO-free DMEM[18]. Cells were cultured for an additional 6 h, after which they were transfected with pCMV-PAD4 plasmid. After overnight incubation, the cells were harvested and cytosolic and histone fractions were prepared as described above. Samples were separated on a single SDS-PAGE, transferred to a PVDF membrane and blotted with the indicated antibodies.

**Micrococcal nuclease (MNase) digestion assay**. The MNase digestion assay was performed according to the previously described method with slight modification[58]. In brief, cell pellets were lysed in a hypotonic buffer (10 mM Tris-HCl pH 7.4, 10 mM KCl, 15 mM $MgCl_2$) on ice for 10 min. Nuclei were pelleted by centrifugation and resuspended in MNase digestion buffer (50 mM Tris-HCl pH 7.9, 5 mM $CaCl_2$) supplemented with RNase and incubated at 37 °C for 30 min. The DNA was then pelleted again by centrifugation and resuspended in MNase digestion buffer supplemented with 100 μg/ml BSA and 40 IU MNase and incubated at room temperature for varying periods of time (0, 5, 10, and 20 min). The MNase reaction was quenched with quenching buffer (0.4 M NaCl, 0.2% (w/v) SDS, 20 mM EDTA) followed by centrifugation. The DNA was extracted and purified by standard procedures, and then analyzed by Tris-Borate-EDTA (TBE) gel electrophoresis.

**Cell viability assay**. Untreated and 24-h PAD4-transfected (WT or C645S) 293 T cells were cultured in a 96-well plate and treated with the indicated MGO concentrations for 12 h. Following the incubation, cell viability was evaluated using

the Cell Counting Kit-8 (CCK-8, Sigma) according to the manufacturer's protocol. The relative cell viabilities were given by detecting the absorbance at 460 nm at each well. Each experiment was performed in triplicate.

**Inhibitor treatment to breast cancer cell lines**. The PAD4 inhibitor GSK484 (Sigma, SML1658; 10 μM) was added to the MCF7 cells' media 6 h prior to adding the corresponding concentrations of MGO. Cells were incubated for additional 12 h after which they were harvested and histones were extracted and analyzed as described above. Samples were separated on a single SDS-PAGE, transferred to a PVDF membrane and blotted with the indicated antibodies.

**Statistics and reproducibility**. For nucleosomal array compaction and cell survival assays, data are presented as the mean ± S.E.M. of three independent experiments. All the western blotting and mass spectrometry data were repeated independently three times with similar results. Statistical analyses were performed in Microsoft Excel, GraphPad Prism 7 with ANOVA, the Student's t-test, or $\chi^2$ test.

**Reporting summary**. Further information on research design is available in the Nature Research Reporting Summary linked to this article.

## Data availability

All the data are available within the article and its Supplementary Information files or from the corresponding author upon reasonable request. The source data underlying Figs. 2, 3 and Supplementary Figs. 2, 3, 4, 7, 8, 10, 12, 13, 14, 15 are provided as a Source Data file. A reporting summary for this article is available as a Supplementary Information file.

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

## Acknowledgements

We would like to thank Prof. Paul Thompson at UMass Medical School for generously sharing the PAD plasmids and the detailed protocol for protein purification. We would also like to thank Prof. Minkui Luo at MSKCC for valuable discussions and critical reading of the manuscript. Work in the David lab is supported by the Josie Robertson Foundation, the Pershing Square Sohn Cancer Research Alliance, the NIH (CCSG core grant P30 CA008748, MSK SPORE P50 CA192937, and R21 DA044767), the Parker Institute for Cancer Immunotherapy (PICI) and the Anna Fuller Trust. In addition, the Davis lab is supported by Mr. William H. Goodwin and Mrs. Alice Goodwin and the Commonwealth Foundation for Cancer Research and the Center for Experimental Therapeutics at Memorial Sloan Kettering Cancer Center. A.O. is supported by the National Science Foundation Graduate Research Fellowship (Grant Number 2016217612) and the Chemical-Biology Interface training grant (NIH T32 GM115327-Tan).

## Author contributions

Q.Z. designed and performed all the experiments with A.O.'s help; Q.Z. and Y.D. analyzed the data; Q.Z. and Y.D. wrote the manuscript with A.O.'s help; Y.D. directed the research.

## Competing interests

The authors declare no competing interests.
