## [Peer Review File · Nature Communications]

Reviewers' comments:

Reviewer #1 (Remarks to the Author):

This is a really interesting study by the David lab showing that PAD4 antagonizes histone methylglyoxal glycation by protecting the reactive sites as well as converting glycated residues to citrulline. The authors also demonstrate that PAD4 and histone citrullination are heightened in breast tumors. The data is clear and convincing and this paper should be accepted as is.

Reviewer #2 (Remarks to the Author):

This manuscript makes several claims that would benefit from rewording and further substantiation.

First, PAD4 is not simply deglycating a histone. It is entirely deiminating the histone, generating a dead-end citrulline at the site of Arg. This is not a pathophysiologically neutral process as the histone can no longer be alkylated or phosphorylated at the Arg residue, which would change chromatin structure and function. This is in contrast with the deglycase DJ-1 that the authors previously claimed to deglycate lysine residues.

Furthermore, the mechanism in Fig. S14A is wrong, the authors propose that the initial aminal adduct somehow becomes an amide (reversibly?) that is then hydrolyzed by PAD4. The existence of an amide should be proven by stability of the glycation product toward hydroxylamine.

The authors also do not discuss/suggest how Lys modifications, which are presumably still occurring in the presence of PAD4, do not directly inhibit chromatin compaction in the context of arrays, which they had claimed in their previous paper in this journal. An experiment demonstrating that the combination of DJ-1 and PAD4 leads to wt array behavior during array aggregation is missing.

There is also no mention of the concomitant DNA modifications by MGO, that would undoubtedly influence chromatin structure and function. Do DNA modifications persist, or are they hydrolyzed over the time-frame of in vitro compaction experiments?

While the observation that PAD4 somehow reverses aminal formation by Arg is interesting, the authors have not made a compelling case that the catalytic activity of PAD4 is at all important for the deglycation reaction. An ITC binding experiment showing that glycated H3 peptide may be bound by catalytically dead PAD4, but is not turned over, would be very revealing in this regard.

Finally, there remains no doubt that excessive PAD4 in the cell would indeed diminish the pool of Arg-containing histones that could react with MGO (an aldehyde that reacts non-selectively with protein/DNA amines). However, PAD4 has many substrates besides the histones. The authors have not demonstrated the transcriptional effects of histone/protein citrullination by performing RNA-seq experiments in their PAD4 transfected human cell lines. Therefore it remains unclear if pre-empting glycation by deimination is harmless to other cellular programs and is indeed a protective pathway under metabolic stress.

Reviewer #3 (Remarks to the Author):

The paper "Protein Arginine Deiminase 4 antagonizes Methylglyoxal-induced Histone Glycation" discusses histone glycation by MGO and the participation of PAD4 in regulating this process, postulating that PAD4 can convert glycated arginine residues into citrulline residues. This process reduces chromatin decompaction - observed as a result of arginine glycation - and prevents these

residues from becoming glycated again. PAD4 was found to be more effective at reducing arginine glycation than the previously identified deglycase DJ-1. Patient tumour samples that exhibit increased glycation also showed increased levels of PAD4 and citrullination compared to non-tumour samples, suggesting a role for this enzyme in the regulation of chromatin structure and function in disease systems.

In general, the manuscript provides significant new biological insight, and the experiments presented proceed in a logical fashion and build the case for PAD4 deglycation of glycated histone arginines, both in model systems and in patient samples. However, a major unresolved question is whether PAD4 is directly utilising the glycated arginine residue as a substrate, or whether PAD4 reacts with free arginine residues that become available as a result of the equilibrium between the glycated and unglycated arginine residues (Figure S14). I therefore feel, that additional experiments would be required to support the authors hypotheses with the necessary biochemical evidence, and therefore this manuscript would have to undergo revisions before it can be published in Nature Communications.

The authors have access to purified PAD4 and peptide substrates, providing an opportunity to characterise this reaction biochemically. The substrate specificity of PAD4 could be investigated by comparing the kinetics of PAD4 citrullination of a glycated substrate with a known reaction specific to free arginine (enzymatic or chemical), also on a glycated substrate. If hydrolysis of the glycated arginine is rate determining (as suggested in Fig. S14B), the reaction rates of the two reactions should be the same, while faster PAD4 kinetics would suggest glycated substrate utilization. Because the argument that PAD4 likely utilises glycated arginine underpins the majority of observations in the report, proving this biochemically would greatly strength the conclusions drawn.

An additional concern is the inconsistency between the dot-blot and mass spectrometry evidence for the citrullination of the H3 terminal peptide presented in Figs. 2D and S5. In 2D, it is clear from the immunostaining that treatment of the H3 peptide with MGO and PAD4 abrogates glycation, presumably by conversion of the glycated arginine residue to a citrulline residue as proposed by the authors. In Fig. S5 however, despite a retention time shift of the major peak, there is no evidence of citrullination in the peptide masses of the peptide (i), citrullinated peptide (ii) or the peptide + MGO and PAD4 (iv). At least at +1 m/z shift would be expected for a single citrullination, and three arginine residues are available on the peptide. The observed masses for each 2+ ion are 1119.9, 1120.0 and 1120.1 m/z, indicating no shift. From Fig. S6, it is clear this shift is detectable on the author's instrumentation, where the expected mass shifts are observed. Based on the MS data in Fig. S5, it appears as if PAD4 is able to reverse the glycation, but leave the arginine free rather than converting it to a citrulline residue. The authors need to reconcile these two observations within their proposed mechanism.

In the discussion, it is noted that histone citrullination also leads to chromatin decompaction, although this is less dramatic than in response to histone glycation. To what extent is citrullination rescuing the disruption caused by glycation, compared to an unmodified histone or a methylated histone? Is this really a rescue or merely damage mitigation? Further discussion would be helpful. The three-way cross talk model proposed by the authors makes sense and is intriguing, but would suggest histone citrullination can accumulate, given this is almost a dead end pathway and the enzyme to reverse this modification is not known.

In summary, the paper presents a logical series of experiments to build the case that PAD4 can reverse histone arginine glycation, through the conversion of glycated arginines to citrulline residues. Proving that PAD4 can directly utilize glycated arginine would greatly strengthen the author's assertion that this interaction is relevant in a disease context.

Minor comments:

Fig. S7 caption includes PAD4, which is not in the figure

Fig. S15 is quite small and difficult to read

Fig. 2c is too small.

Reviewer #1:

This is a really interesting study by the David lab showing that PAD4 antagonizes histone methylglyoxal glycation by protecting the reactive sites as well as converting glycated residues to citrulline. The authors also demonstrate that PAD4 and histone citrullination are heightened in breast tumors. The data is clear and convincing and this paper should be accepted as is.

We are pleased to see that Reviewer 1 is satisfied with our manuscript and appreciate their efforts to review our work.

Reviewer #2:

This manuscript makes several claims that would benefit from rewording and further substantiation.

- 1. First, PAD4 is not simply deglycating a histone. It is entirely deiminating the histone, generating a dead-end citrulline at the site of Arg. This is not a pathophysiologically neutral process as the histone can no longer be alkylated or phosphorylated at the Arg residue, which would change chromatin structure and function. This is in contrast with the deglycase DJ-1 that the authors previously claimed to deglycate lysine residues.*

We thank the reviewer for bringing up this important point. PAD4 indeed converts unmodified and modified arginine residues (*i.e.*, methylated and MGO-glycated) to citrulline, so in our manuscript, unlike DJ-1, PAD4 is referred to as a “rewriter” instead of an “eraser”. The histone citrullination activity of PAD4 is a reported pathophysiological neutral process (*Curr. Drug Targets*, 2015, 16, 700-710; *Front Immunol.*, 2012, 3:307), although there is no known eraser enzyme converting histone citrulline back to arginine. To stress this important point in accordance with the reviewer’s suggestion, we have expanded the discussion in the revised manuscript, which now reads: “Our data (Figures 2 and S15) support a dual-functional model whereby PAD4 is capable of not only protecting arginine via deamination but rewriting MGO-adduct intermediates into citrulline, which is currently speculated to be a terminal product (Figures 1, S16, and S17). There are several implications for this newly identified rewriting function of PAD4 on MGO-glycated histones.....”

- 2. Furthermore, the mechanism in Fig. S14A is wrong, the authors propose that the initial aminor adduct somehow becomes an amide (reversibly?) that is then hydrolyzed by PAD4. The existence of an amide should be proven by stability of the glycation product toward hydroxylamine.*

We agree with the reviewer that the proposed mechanism for PAD4 activity is not trivial, but is inspired by the most widely accepted deglycation mechanism of DJ-1 (*Hum. Mol. Genet.*, 2012, 21, 3215-3225; *J. Biol. Chem.*, 2015, 290, 1885-1897; *Science*, 2017, 357, 208-211). In the proposed mechanism for DJ-1, depicted in the figure below, it facilitates the isomerization of aminocarbinal to amide, where the two intermediates are in a dynamic equilibrium. The subsequent thioester formation and DJ-1-mediated hydrolysis push the equilibrium forward to complete the deglycation reaction. However, since this proposed mechanism of PAD4 (old Fig. S14A) is less trivial than the one shown in Figure S14C (based on the current understandings of PAD4 biochemistry), we deleted

it from the revised SI. In addition, based on the results of the experiments suggested by Reviewer 3, we now have also excluded the mechanism proposed in Figure S14B. The final proposed mechanism for PAD4 deglycase activity is now summarized in revised Figure S16.

Proposed mechanism of DJ-1's deglycase activity

3. *The authors also do not discuss/suggest how Lys modifications, which are presumably still occurring in the presence of PAD4, do not directly inhibit chromatin compaction in the context of arrays, which they had claimed in their previous paper in this journal. An experiment demonstrating that the combination of DJ-1 and PAD4 leads to wt array behavior during array aggregation is missing.*

We appreciate the reviewer's experimental suggestion. The combined treatment of DJ-1 and PAD4 of MGO-glycated arrays was performed and the corresponding data added to the new **Figure S8b** in our revised SI. These data show that the co-treatment of DJ-1 and PAD4 cannot rescue the glycated arrays back to the original compaction states, due to the conversion of glycated arginine to citrulline by PAD4.

Combined treatment of DJ-1 and PAD4 of MGO-glycated arrays

4. *There is also no mention of the concomitant DNA modifications by MGO, that would undoubtedly influence chromatin structure and function. Do DNA modifications persist, or are they hydrolyzed over the time-frame of in vitro compaction experiments?*

We thank the reviewer for bringing up this important point. Indeed, in our previous paper we have discussed DNA MGO-glycation and its impact on chromatin as a transcriptional template (Figures 3b, 3c, and 4b in *Nat. Commun.*, 2019, 10:1289). This is primarily due to the fact that DJ-1 is also a reported DNA deglycase (*Science*, 2017, 357, 208-211). However, PAD4 is not a DNA deglycase based on its catalytic mechanism, as it can only recognize guanidino groups of specific targets (histones H3, H4, H1, and some non-histone proteins). To clarify this point in the manuscript we have added discussion regarding the substrate differences between DJ-1 and PAD4, which now reads: “The results we present here suggest that this could be due to the fact that DJ-1 can only erase early stage glycation products (including both N-glycated proteins and nucleotides) (18, 23-25), while PAD4 can specifically remove glycation modifications from arginine side-chains in a subset of proteins (including in the N-terminal residues of histones) (12), which are key epigenetic regulators, by converting them to citrullines. Furthermore, other PAD enzymes, as well as PAD4 itself, may target glycation adducts in other substrates although this remains to be determined (12).”

5. *While the observation that PAD4 somehow reverses aminal formation by Arg is interesting, the authors have not made a compelling case that the catalytic activity of PAD4 is at all important for the deglycation reaction. An ITC binding experiment showing that glycated H3 peptide may be bound by catalytically dead PAD4, but is not turned over, would be very revealing in this regard.*

The reviewer suggests an invaluable experiment that can provide further evidence for the direct binding between glycated H3 peptides and catalytically dead PAD4. To test the *in vitro* and *in cellulo* binding between the MGO-glycated H3 peptide and catalytically-dead PAD4 from both heterologous expression and 293T cell lysates, we performed a pull-down assay using the biotin-containing H3 peptide (**new Figure S12**). The results of the experiment, shown below, demonstrate that both MGO-modified and unmodified H3 N-terminal tails can bind catalytically-dead PAD4 (PAD4-C645S) *in vitro* and *in vivo* without further conversion.

Immunoprecipitation (IP) and dot blot analyses of H3 N-terminal peptide (residues 1-18) binding to catalytically dead PAD4 mutant (PAD4-C645S)

6. *Finally, there remains no doubt that excessive PAD4 in the cell would indeed diminish the pool of Arg-containing histones that could react with MGO (an aldehyde that reacts non-selectively with protein/DNA amines). However, PAD4 has many substrates besides the histones. The authors have not demonstrated the transcriptional effects of histone/protein citrullination by performing RNA-seq experiments in their PAD4 transfected human cell lines. Therefore it remains unclear if pre-empting glycation by deimination is harmless to other cellular programs and is indeed a protective pathway under metabolic stress.*

We completely agree with the reviewer that PAD4 has many substrates beyond histones (such as transcriptional effectors), and that PAD4 activity induces changes in cellular transcription and overall state. However, this point had already been demonstrated previously using RNA-Seq as well as proteomic experiments in PAD4 transfected, inhibitor-treated or knockdown cell lines (*JCI Insight*, 2018, 3, e124729; *Cell Rep.*, 2018, 22, 1473-1483; *Mol. Cell Biol.*, 2018, 38, e00084-18). Moreover, RNA-Seq analysis of MGO-treated cell lines has shown significant effect on the cellular transcriptome too (*BioChip J.*, 2011, 5, 220; *Proc. Natl Acad. Sci. USA.*, 2018, 115, 9228-9233). We have performed complementary assays (chromatin MNase digestion) to show the implications of PAD4 on chromatin architecture. However, the key discovery in our manuscript is that PAD4 antagonizes histone MGO-glycation by protecting unmodified arginine residues via deimination and rewriting MGO-modified arginine into citrulline. This newly identified function will not have additional effects besides the known pathophysiological role of PAD4 in epigenetics (*i.e.*, induction of heterochromatin decondensation; *Front Immunol.*, 2012, 3:307). To stress this point, we have added a discussion and several citations regarding the overall epigenetic function of the writer enzyme PAD4, which now reads: “Overall, these findings expand our understandings of PAD4 biochemistry and its pathophysiological function in human health (46-50).”

Reviewer #3:

A major unresolved question is whether PAD4 is directly utilising the glycated arginine residue as a substrate, or whether PAD4 reacts with free arginine residues that become available as a result of the equilibrium between the glycated and unglycated arginine residues (Figure S14). I therefore feel, that additional experiments would be required to support the authors hypotheses with the necessary biochemical evidence, and therefore this manuscript would have to undergo revisions before it can be published in Nature Communications.

1. *The authors have access to purified PAD4 and peptide substrates, providing an opportunity to characterise this reaction biochemically. The substrate specificity of PAD4 could be investigated by comparing the kinetics of PAD4 citrullination of a glycated substrate with a known reaction specific to free arginine (enzymatic or chemical), also on a glycated substrate. If hydrolysis of the glycated arginine is rate determining (as suggested in Fig. S14B), the reaction rates of the two reactions should be the same, while faster PAD4 kinetics would suggest glycated substrate utilization. Because the argument that PAD4 likely utilises glycated arginine underpins the majority of observations in the report, proving this biochemically would greatly*

strengthen the conclusions drawn.

We appreciate the reviewer's perceptive observations and invaluable experimental suggestion to strengthen our manuscript. To address this concern, we first attempted to perform MGO-glycation with free arginine as a substrate, however, arginine reacted so quickly with MGO, forming a series of heterogeneous adducts, that we were unable to separate for the further quantitative LCMS-based analysis. Next, we turned to PRMT1, which specifically methylates H4R3 to generate an asymmetric H4R3me2. While we demonstrated the rewriting activity of PAD4 on MGO-glycated H4R3 with a 1-6aa H4 peptide (Figures 2E and S6), this peptide is too short to serve as a proper substrate for PRMT1. The reported suitable H4 peptides, such as H4 (1-21aa) (*Biochemistry*, 2007, 46, 13370-13381), contain too many MGO-glycation sites (3 arginines and 4 lysines) to be utilized in making homogeneous MGO-glycated products or for quantitative LCMS-based analysis. We thus decided to use the optimized substrate, full-length H4, and apply two site-specific antibodies, anti-H4Cit3 (MilliporeSigma, 07-596) and anti-H4R3me2 (Abcam, ab194683), to demonstrate PAD4's direct utilization of glycated H4, compared with protecting histone arginines by shifting the chemical equilibrium. We blotted MGO-glycation of H4 before and after enzyme-treatment (PAD4 or PRMT1), followed by a western blot analysis. The results (Figure S7) provide direct evidence that PAD4 utilizes glycated histones as substrates while PRMT1 does not, excluding the proposed mechanism in the old Figure S14B. The revised proposed mechanism of PAD4's dual function for antagonizing histone MGO-glycation is now summarized in the revised Figure S16.

Comparison of PAD4 and PRMT1 activities on MGO glycated and full-length histone H4

2. *An additional concern is the inconsistency between the dot-blot and mass spectrometry evidence for the citrullination of the H3 terminal peptide presented in Figs. 2D and S5. In 2D, it is clear from the immunostaining that treatment of the H3 peptide with MGO and PAD4 abrogates glycation, presumably by conversion of the glycated arginine residue to a citrulline residue as proposed by the authors. In Fig. S5 however, despite a retention time shift of the major peak, there is no evidence of citrullination in the peptide masses of the peptide (i), citrullinated peptide (ii) or the peptide + MGO and PAD4 (iv). At least at +1 m/z shift would be expected for*

a single citrullination, and three arginine residues are available on the peptide. The observed masses for each 2+ ion are 1119.9, 1120.0 and 1120.1 m/z , indicating no shift. From Fig. S6, it is clear this shift is detectable on the author's instrumentation, where the expected mass shifts are observed. Based on the MS data in Fig. S5, it appears as if PAD4 is able to reverse the glycation, but leave the arginine free rather than converting it to a citrulline residue. The authors need to reconcile these two observations within their proposed mechanism.

We thank the reviewer for taking a thorough look at our data, about which, we originally had similar concern. However, we found that the difference in mass is due to the averaging deconvolution of our ESI-LCMS and its limited resolution. The longer the peptide we separate and analyze on the instrument, the larger the mass shift we observe following deconvolution. For example, when we perform mass deconvolution of full-length histone proteins, we observe ± 2 Da of the calculated mass. For the synthetic 6-amino acid H4 peptide, the MS shift of H4-Cit3(1-6) to H4-R3(1-6) is 0.8 Da (Figure 2e), which is also less than 1 Da. For the 19-amino acid H3 peptide, the MS shifts are 0.2 and 0.4 Da after citrullination (**Figure S5**), which is not beyond our expected difference. Similar observations were reported by other labs, where mono-citrullinated peptides have the same mass as protonated unmodified arginine peptide by LC-MS (see Figure S1 in *JACS*, 2012, 134, 17015-17018). However, to address the reviewer's concerns, we performed a similar reaction in $H_2^{18}O$ that amplifies the mass shift and the corresponding result are presented in revised **Figure S5**.

LC-MS analysis of MGO induced glycation and PAD4 mediated deglycation of H3 N-terminal peptide substrate

- In the discussion, it is noted that histone citrullination also leads to chromatin decompaction, although this is less dramatic than in response to histone glycation. To what extent is citrullination rescuing the disruption caused by glycation, compared to an unmodified histone or a methylated histone? Is this really a rescue or merely damage mitigation? Further discussion would be helpful. The three-way cross talk model proposed by the authors makes sense and is intriguing, but would suggest histone citrullination can accumulate, given this is almost a dead end pathway and the enzyme to reverse this modification is not known.*

We thank the reviewer for their helpful advice. In our previous paper (Figure 4g in *Nat. Commun.*, 2019, 10:1289), we have shown that the major MGO-glycation damage on chromatin is the formation of crosslinks and AGEs leading the transcription block and cell death, while mild histone glycation is beneficial to gene transcription and cell vitality. Since citrulline is, to the best of our knowledge, a terminal modification as it prevents any other adducts (including glycation and crosslinking), it might be considered more of a “protection” rather than “rescue” pathway. According to Reviewers 2 and 3’s suggestions, we have added a discussion regarding this role of histone citrulline in the revised manuscript, which now reads: “Our data (Figures 2 and S15) support a dual-functional model whereby PAD4 is capable of not only protecting arginine via deamination but rewriting MGO-adduct intermediates into citrulline, which is currently speculated to be a terminal product (Figures 1, S16, and S17). There are several implications for this newly identified rewriting function of PAD4 on MGO-glycated histones.....”

- Minor comments:

Fig. S7 caption includes PAD4, which is not in the figure

Fig. S15 is quite small and difficult to read

Fig. 2c is too small.

We thank the reviewer for their careful reading of our manuscript – we have revised these figures according to the reviewer’s comments.

REVIEWERS' COMMENTS:

Reviewer #2 (Remarks to the Author):

In response to the reviewers queries the authors revised several elements. Unfortunately, several key issues with this manuscript that reduce enthusiasm are:

- 1) There is no mechanistic basis for suggesting that PAD4 would remove MGO from Arg side-chains similar to DJ-1. Thus, the mechanism is only speculative and has little meaning for PAD4 which is a different enzyme than DJ-1 with poor sequence homology. Changing the labels in the mechanism from PAD4 to DJ1 does not make it correct for PAD4. Even the suggestions for DJ-1 are speculative (in the JBC paper, the Science paper has no mechanism relevant to an amide hydrolysis by DJ-1)
- 2) Another key issue, is that the authors have now shown that the combination of DJ-1 and PAD4 acting together still do not rescue chromatin compaction (new Figure S8b). This is worrisome and confirms that the proposed mechanism is unlikely to be physiologically neutral and should lead to untimely decompaction of chromatin which is an undesirable fate.
- 3) In fact, the authors cite the paper in *Front Immunol.* 2012; 3: 307 as evidence that citullination is pathophysiologically neutral. However, the manuscript clearly states that, "Strikingly, transient transfection of PAD4 for 36 h induced U2OS cells to rupture and release extensive web-like chromatin fibers into the extracellular space". Furthermore, "Histone citrullination induces HP1 β dissociation and heterochromatin decondensation" is a bad outcome.

While the initial observation made by the authors is interesting, overall, the current manuscript does not rise to a relevant level for publication in *Nat. Comm.* I would recommend publication in a more specialized journal where much of the PAD4 literature is already seen, such as *ACS Chemical Biology* or *ACS Biochemistry*.

Reviewer #3 (Remarks to the Author):

The authors have addressed all my questions and I am happy to fully support publication of the manuscript as is.

Reviewer #2 (Remarks to the Author):

In response to the reviewers queries the authors revised several elements.

We thank the reviewer for acknowledging our efforts to revise the paper during these challenging times.

Unfortunately, several key issues with this manuscript that reduce enthusiasm are:

1) There is no mechanistic basis for suggesting that PAD4 would remove MGO from Arg side-chains similar to DJ-1. Thus, the mechanism is only speculative and has little meaning for PAD4 which is a different enzyme than DJ-1 with poor sequence homology. Changing the labels in the mechanism from PAD4 to DJ1 does not make it correct for PAD4. Even the suggestions for DJ-1 are speculative (in the JBC paper, the Science paper has no mechanism relevant to an amide hydrolysis by DJ-1)

We apologize for this misunderstanding, as we never intended to suggest that the mechanisms for PAD4's rewriting and DJ-1's deglycation activities are similar. In fact, we hoped to stress the differences between the two enzymes in their mechanisms of action, substrates, and products throughout the manuscript. Briefly, while the proposed mechanism for DJ-1's deglycation relies on a mercapto adduct between the catalytic residue of the enzyme (Cys 106) and the dicarbonyl adduct followed by the formation of DJ-1-lactate thioester intermediate, the mechanism for PAD4's rewriting relies on a thioether intermediate, resulting in the deamination and subsequent nucleophilic attack of a water molecule to generate the citrullinated arginine. We provide multiple evidence for the PAD4's mechanism *in vitro*, by western blot and RP-HPLC analyses as well as mass spectrometry using ¹⁸O labeled water. This mechanism is reinforced by our pulse-chase as well as genetic and pharmacological perturbations *in cellulo* experiments. Together, our results strongly support this mechanism and distinguish it from the one proposed for DJ-1.

To further clarify this in the discussion we now added the following explanation:

“DJ-1 and PAD4 also work through distinct catalytic mechanisms. While DJ-1 erases early MGO-glycation adducts from both lysines and arginines, PAD4 is only active on arginine residues, that are more reactive towards MGO relative to lysines (20, 21). However, PAD4 is superior due to its rewriting activity, that is, it removes the glycation adduct from arginine and protects it from further damage by converting it to citrulline.”

2) Another key issue, is that the authors have now shown that the combination of DJ-1 and PAD4 acting together still do not rescue chromatin compaction (new Figure S8b). This is worrisome and confirms that the proposed mechanism is unlikely to be physiologically neutral and should lead to untimely decompaction of chromatin which is an undesirable fate.

We believe that the source of this confusion may be in the interpretation of our *in vitro* compaction data in Figure S8. In this experiment, early glycation induces fiber decompaction and DJ-1 converts it back to untreated arrays compaction state due to its deglycation activity. However, PAD4 is a rewriter that converts the glycated arginine to citrulline (and not back to arginine). Thus, we cannot and would not compare the synergistic activity of PAD4 and DJ-1 to untreated arrays. By definition, after PAD4 treatment, the arrays will be decorated with arginine citullination, which has been shown to induce chromatin decompaction (*J. Cell Biol.*, 2009, 184, 205-213; *Nature*, 2014, 507, 104-108.). This is precisely in line with our compaction data, where PAD4-treated arrays (either glycated or non-glycated) are slightly less compacted than untreated arrays.

To further clarify this in the discussion we now added the following explanation:

“Both histone glycation at its early stages (18) and citrullination (7, 11) induce chromatin decompaction. Indeed, the synergistic activity between PAD4 and DJ-1 leaves the treated glycated and non-glycated chromatin less compacted, with the newly added citrullination inducing chromatin fiber decompaction compared with untreated one (Figures 2c, S8 and S11).”

3) In fact, the authors cite the paper in *Front Immunol.* 2012; 3: 307 as evidence that citullination is pathophysiologically neutral. However, the manuscript clearly states that, “Strikingly, transient transfection of PAD4 for 36 h induced U2OS cells to rupture and release extensive web-like chromatin fibers into the extracellular space”. Futhermore, “Histone citrullination induces HP1 β dissociation and heterochromatin decondensation” is a bad outcome.

We cited the *Frontiers in Immunology* paper as the authors demonstrate the effect PAD4 overexpression has on cellular phenotype through their proteomics and RNA-seq analyses. Their results suggest that PAD4 overexpression is detrimental to cells, as it participates in global chromatin decompaction that produces NET-like extracellular structures. While this report describes the role PAD4 has in neutrophils, other reports, as well as ours, suggest PAD4 is an oncoprotein (*Cancer Res.*, 2019, 79, 1274-1284; *Biochem. Res. Int.* 2012, 895343; *BMC Cancer*, 2009, 9, 40). While the precise mechanism for PAD4’s oncogenic activity is poorly understood, oncoproteins activity in general often induces “bad outcomes” (*Oncogene*, 1999, 18, 2281–2290; *J. Endod.*, 2016, 42, 575-583.), but provides an advantage to the cells that survive, subsequently nucleating cancer tumor formation (*Oncogene*, 2016, 35, 4036–4047; *Biochim. Biophys. Acta.*, 1994, 1226, 89-96.). Indeed, we show the overexpression of PAD4 as well as the resulting increased levels of H3 citrullination in our breast-cancer patients’ tumors, which supports the pathophysiological phenotype.

To further clarify this in the discussion we now added the following explanation:

“...both DJ-1 and PAD4 are proposed oncoproteins and targets of cancer therapy although the mechanisms are not fully understood (13-16, 28). Indeed, we have shown

that breast cancer patient samples contain massive overexpression of both DJ-1 (18) and PAD4 (Figure 3).”

Reviewer #3 (Remarks to the Author):

The authors have addressed all my questions and I am happy to fully support publication of the manuscript as is.

We thank the reviewer for their efforts and insightful comments that improved the quality of our manuscript.